# Scorpion Venom Antimicrobial Peptide Derivative BmKn2-T5 Inhibits Enterovirus 71 in the Early Stages of the Viral Life Cycle *In Vitro*

**DOI:** 10.3390/biom14050545

**Published:** 2024-05-01

**Authors:** Zhiqiang Xia, Huijuan Wang, Weilie Chen, Aili Wang, Zhijian Cao

**Affiliations:** 1Center for Evolution and Conservation Biology, Southern Marine Science and Engineering Guangdong Laboratory (Guangzhou), Guangzhou 511458, China; 2016202040013@whu.edu.cn; 2State Key Laboratory of Virology, College of Life Sciences, Wuhan University, Wuhan 430072, China; 2016282040175@whu.edu.cn; 3School of Biological and Food Processing Engineering, Huanghuai University, Zhumadian 463000, China; 4Shenzhen Research Institute, Wuhan University, Shenzhen 518057, China; 5Institute of Infectious Diseases, Guangzhou Eighth People’s Hospital, Guangzhou Medical University, Guangzhou 510060, China; gz8hcwl@gzhmu.edu.cn

**Keywords:** scorpion venom peptides, antimicrobial peptides, BmKn2-T5, EV71, antiviral activity

## Abstract

Enterovirus 71 (EV71), a typical representative of unenveloped RNA viruses, is the main pathogenic factor responsible for hand, foot, and mouth disease (HFMD) in infants. This disease seriously threatens the health and lives of humans worldwide, especially in the Asia–Pacific region. Numerous animal antimicrobial peptides have been found with protective functions against viruses, bacteria, fungi, parasites, and other pathogens, but there are few studies on the use of scorpion-derived antimicrobial peptides against unenveloped viruses. Here, we investigated the antiviral activities of scorpion venom antimicrobial peptide BmKn2 and five derivatives, finding that BmKn2 and its derivative BmKn2-T5 exhibit a significant inhibitory effect on EV71. Although both peptides exhibit characteristics typical of amphiphilic α-helices in terms of their secondary structure, BmKn2-T5 displayed lower cellular cytotoxicity than BmKn2. BmKn2-T5 was further found to inhibit EV71 in a dose-dependent manner in vitro. Moreover, time-of-drug-addition experiments showed that BmKn2-T5 mainly restricts EV71, but not its virion or replication, at the early stages of the viral cycle. Interestingly, BmKn2-T5 was also found to suppress the replication of the enveloped viruses DENV, ZIKV, and HSV-1 in the early stages of the viral cycle, which suggests they may share a common early infection step with EV71. Together, the results of our study identified that the scorpion-derived antimicrobial peptide BmKn2-T5 showed valuable antiviral properties against EV71 *in vitro*, but also against other enveloped viruses, making it a potential new candidate therapeutic molecule.

## 1. Introduction

Enterovirus 71 (EV71), belonging to the genus Enterovirus in the family of Picornavirridae, is a single-stranded, non-enveloped RNA virus of about 30 nm in diameter [1]. Its virion is protected by an icosahedral capsid consisting of VP1, VP2, and VP3, and a positive-sense RNA 7.5 kb nucleotides in length is located on the internal part of the capsid [2]. As the main pathogen of hand, foot, and mouth disease (HFMD), EV71 was first isolated from meningitis patients in California in 1969, and it mainly infects infants and children under 6 years of age [3,4]. Generally, the clinical symptoms caused by EV71 infection are mild and self-limiting, including fever and painful blister-like sores. In particular, due to its high neurotropism feature, it can also cause severe neurological complications, such as brainstem encephalitis, aseptic meningitis, acute flaccid paralysis, pulmonary complications, and even death [5,6]. In the last few decades, EV71 has gradually become a significant threat to human health throughout the world, especially in Australia, Singapore, Japan, China, Malaysia, and other countries of Asia–Pacific, resulting in hospitalization and mortality rates ranging from 5% to 19% in children [7,8,9,10]. Most studies to date have focused on the identification of novel host factors, the IFN-mediated antiviral response, the discovery or development of effective antivirals, and the elucidation of host–pathogen interactions during EV71 infection [3,11,12,13]. For example, EV71 (2Apro) has been found to upregulate the expression and secretion of LDL-receptor-related protein-associated protein 1 (LRPAP1), and the N-terminus of secreted LRPAP1 can bind with the extracellular domain of IFNAR1, triggering the receptor’s ubiquitination and degradation and promoting virus infection both in vitro and in vivo [14]. Additionally, two cathelicidin-derived peptides, LL-18 and FF-18, could directly bind to the EV71 virus particles and block virus–receptor interactions, showing more potent action against EV71 infection [13]. Although vaccine protection remains the primary measure for addressing EV71 infection, drug treatment for severe infections is still of the utmost importance [15]. However, no specific drugs against EV71 have been approved for clinical therapeutics, and supportive therapy is only able to relieve symptoms at present. Therefore, it is necessary to find an effective antiviral drug for the treatment of EV71 infection.

As important effectors of the innate immune system in fighting against invasion by pathogens such as bacteria, fungi, viruses, and other microorganisms, antimicrobial peptides (AMPs) are a type of small-molecule peptide widely distributed in most living organisms. Due to their potent antimicrobial activity and unique antimicrobial mechanisms, they have become a hot spot of medical research and have the potential to be developed as therapeutic agents for treating drug-resistant bacteria [16,17]. As of December 2023, a total of 3940 AMPs were included in the Antimicrobial Peptide Database (http://aps.unmc.edu/AP, accessed on 31 December 2023), and most were identified from animals, especially insects. As one of the most ancient chelicerates that have evolved over an estimated 430 million years, scorpions belong to the phylum Arthropoda, class Arachnida, and order Scorpionida and have traditionally been used for medicinal purposes in some Asian and African countries [18,19,20]. Because the physical process of venom injection may damage the caudal ganglia, resulting in further pathogen infection, it is expected that scorpions would possess an antimicrobial response system comprising unique mechanisms, such as AMPs, to protect themselves from microbe invasion [21,22].

Through the structural and functional characterization of scorpion venom components, a significant percentage of scorpion-derived AMPs have been identified, showing potential pharmacological activities in antibacterial, antifungal, and antiparasitic aspects. Currently, more than 25 short-chain and 4 long-chain AMPs with antibacterial properties have been isolated from different scorpion species, including *Mesobuthus martensii*, *Androctonus aeneas*, *Tityus serrulatus*, *Opisthacanthus madagascarieni*, and *Heterometrus spinifer* [23,24,25,26]. For instance, AaeAP1 and AaeAP2, isolated from the North African scorpion *A. aeneas*, contain 17 amino acids without disulfide bridges and exhibit more selective growth-inhibitory activities against *Staphylococcus aureus* (16 mg/L) than against *Escherichia coli* (512 mg/L) [25]. In addition, 19 scorpion-derived peptides have been confirmed to exert significant antifungal activity and 8 peptides with antiparasitic activity have been reported to date [22,23,27]. Interestingly, four peptide analogs of stigmurin from the scorpion T. stigmurus, named StigA6, StigA16, StigA25, and StigA31, not only exhibit antibacterial and antifungal effects superior to those of the native peptide but also efficiently inhibit the growth of epimastigote forms of *Trypanosoma cruzi*, suggesting that AMPs may serve as potential broad-spectrum therapeutic agents in resistance against foreign invading microorganisms [28,29]. Although 12 scorpion-derived peptides have been confirmed to exert inhibitory activity against viral infection, including hepatitis B virus (HBV), hepatitis C virus (HCV), herpes simplex virus (HSV), and human immunodeficiency virus (HIV), the development and optimization of scorpion venom peptides with promising antiviral activity against different viral families is still worth further exploration [23,30,31,32].

In a previous study by our group, we demonstrated that BmKn2, isolated from the scorpion *M. martensii*, and its derivative BmKn2-7 showed significant inhibitory activity against both Gram-positive and Gram-negative bacteria. Moreover, BmKn2 and its derivative BmKn2-7 could inhibit HIV-1 via direct interaction with viral particles, indicating that these AMPs can be developed as potential antiviral agents [33]. In this study, we find that BmKn2 and its derivative BmKn2-T5 have a typical amphiphilic α-helical structure and exhibit significant inhibitory activity against unenveloped viruses EV71. Compared with BmKn2, BmKn2-T5 demonstrates lower cytotoxicity in RD cells and dose-dependently inhibits EV71 infection at noncytotoxic concentrations. In addition, BmKn2-T5 mainly restricts EV71 infection, but not its virion or replication, at the early stage. Finally, we demonstrated that BmKn2-T5 surprisingly showed activity against DENV, ZIKV, and HSV-1, suggesting that it might serve as a broad-spectrum antiviral agent and lead to the discovery of new candidates for modern drug development.

## 2. Materials and Methods

### 2.1. Cells and Viruses

Human rhabdomyosarcoma (RD) cells were kindly donated by Prof. Wanhong Liu from the School of Basic Medicine, Wuhan University in China, and cultured in Minimum Essential Medium (MEM, Gibco-Invitrogen, New York, NY, USA) supplemented with 10% fetal bovine serum (Gibco-Invitrogen) and 1% penicillin/streptomycin at 37 °C in a humidified 5% CO_2_ incubator. The African green monkey kidney cells (Vero) stored in our laboratory were cultured in Dulbecco’s modified Eagle’s medium (DMEM, Gibco-Invitrogen) with the same concentration of fetal bovine serum or penicillin/streptomycin. The international standard strain EV71 BrCr was also provided by Prof. Wanhong Liu. The DENV2 TSV01 strain was kindly provided by Dr. Bo Zhang from the Wuhan Institute of Virology, Chinese Academy of Sciences. cDNA plasmid for the ZIKV Puerto Rico strain (PRVABC59) was kindly provided by Dr. Ren Sun and Dr. Danyang Gong at University of California, Los Angeles. Herpes simplex virus type I (HSV-1) was stored in our laboratory.

### 2.2. Reagents and Antibodies

The scorpion venom-derived antimicrobial peptide BmKn2 and its five derivatives were all synthesized by GL Biochem (Shanghai, China) with purity greater than 95%. TRIzol reagent (#9108), used for RNA extraction, and the RevertAid First Strand cDNA Synthesis Kit (#K1622) were purchased from Takara and ThermoFisher Scientific, respectively. EV71 VP1 monoclonal antibody produced in rabbits was ordered from Abcam (#ab308205, Cambridge, UK). The glyceraldehyde-3-phosphate dehydrogenase (GAPDH) mouse antibody was purchased from the ProteinTech Group (#60004-1-Ig, Wuhan, China).

### 2.3. MTS Assay

The cytotoxicity of BmKn2 and BmKn2-T5 in RD cells was assessed using the MTS assay (#G1112, Promega, Promega, Madison, WI, USA). Briefly, RD cells were seeded into 96-well plates at a density of 5 × 10^3^ for 12 h, followed by incubation with various concentrations of each peptide when the cells reached 80% confluency. After 48 h of incubation, the cell culture medium was removed, and the cells were cultured in 100 μL fresh medium mixed with 20 µL of MTS reagent per well for 40 min at 37 °C. The absorbance at 490 nm wavelength was measured using the EL × 800 Absorbance Microplate Reader (Biotek, Sinergy, HT, Winooski, VT, USA).

### 2.4. Circular Dichroism (CD) Analysis

CD spectra are an excellent tool used to estimate the secondary structures of unknown peptides or proteins [34,35]. To further determine the structural characteristics of BmKn2 and its derivative BmKn2-T5, CD analysis of them was performed using a spectropolarimeter (JASCO J-810, JASCO International Co., Ltd., Tokyo, Japan). At first, two peptides were, respectively, resuspended in ultrapure water, 30% 2,2,2-trifluoroethanol (TFE, mimicking the hydrophobic environment of the microbial membrane), or 70% TFE to a final concentration of 150 µg/mL. Then, these samples were placed into a 0.1 cm path-length quartz cuvette at 25 °C, and the data were recorded in the range of 185–250 nm at a scan rate of 50 nm/min. For each peptide, spectra were collected from three separate recordings which were baseline-corrected by subtracting the solvents in identical conditions from the spectra. Finally, the acquired CD spectra were converted to the mean residue ellipticity through the following equation calculated using CD deconvolution software (Spectra Manager™II): θM = (θobs × 1000)/(c × l × n). Here, θM is the mean residue ellipticity (deg cm2 dmol-1), θobs is the observed ellipticity corrected for the buffer at a given wavelength (mdeg), c is the peptide concentration (mM), l is the path length (mm), and n is the number of amino acids.

### 2.5. qRT-PCR

To quantify the expression of different genes at the transcriptional level, intracellular total RNA was extracted using TRIzol reagent according to the manufacturer’s instructions. Then, a 0.5–1 μg aliquot of total RNA was used as a template for cDNA synthesis using the RevertAid First Strand cDNA Synthesis Kit, and intracellular EV71, DENV, and ZIKV RNAs were detected through quantitative real-time PCR (qRT-PCR) using the Bestar SYBR Green qPCR master mix reagent (DBI^®^ Bioscience, Ludwigshafen, Germany). Each qRT-PCR contained 10 μL of the mixed reagent, 0.5 μL of forward and reverse primers, 1 μL of template, and 8 μL of ddH_2_O, making a total volume of 20 μL. All the gene-specific qRT-PCR primers used are listed in Table 1, and the gene copy numbers were determined using a 7500 Real-Time PCR system (Applied Biosystems, Waltham, MA, USA) and the comparative method (∆∆CT). Experiments were undertaken independently in triplicate with duplicate real-time PCRs, and all data are presented as the means ± SD.

### 2.6. Western Blotting

Cultured cells were first washed with ice-cold phosphate-buffered saline (PBS) and then were lysed in ice-cold RIPA lysis buffer (ThermoFisher Scientific, Waltham, MA, USA) supplemented with protease and phosphatase inhibitor cocktail. The lysates were then heated in a boiling water bath for 20 min, and the concentrations of total protein were determined using the Pierce™ BCA Protein Assay Kit (ThermoFisher Scientific). After total protein separation on a 10% SDS-PAGE gel, all component proteins were then transferred from the gel onto a nitrocellulose membrane (Millipore, Billerica, MA, USA). Furthermore, the membrane was blocked by incubating it first with 5% skim milk at room temperature and then with the primary antibody at a temperature of 4 °C overnight. Anti-EV71 VP1 antibody was used at 1:2000 dilution, and anti-GAPDH was used at 1:10,000 dilution as a loading control for quantification normalization. Subsequently, membranes were exposed to HRP-conjugated anti-mouse or anti-rabbit secondary antibodies at room temperature for 2 h, and the results were then visualized using ClarityTM Western ECL Substrates (BIO-RAD, #1705060) with Fuji medical X-ray film.

### 2.7. Time of Addition Assay

To determine the steps and mechanism of action affected by BmKn2-T5, it was added to virus or cells at different times as previously described [36,37]. For free virion, a tube containing EV71 particles and 10 μg/mL BmKn2-T5 mixture was first incubated for 1 h at 37 °C, and the mixture was then diluted and added to cells for 1 h. After washing off the unabsorbed virus with PBS, the cells were further cultured for 12 h, and intracellular RNA was detected using qRT-PCR. For attachment, RD cells infected with EV71 at an MOI of 10 were incubated with 10 μg/mL BmKn2-T5 at 4 °C for 1 h, and the cells were then further incubated for 12 h at 37 °C followed by washing off the unabsorbed virus using PBS. For entry, RD cells were infected with EV71 at an MOI of 10 for 1 h at 4 °C, and the mixture was then treated with 10 μg/mL BmKn2-T5 for 1 h at 37 °C after washing off the unabsorbed virus. Finally, cells were further cultured for 12 h after washing off the unabsorbed virus again, and intracellular RNA was detected using qRT-PCR. For replication, RD cells infected with EV71 at an MOI of 10 were incubated with 10 μg/mL BmKn2-T5 at 37 °C for 1 h, and the cells were then further incubated at 12 h at 37 °C followed by washing off the unabsorbed virus using PBS. For the time of addition assay, RD cells were first infected with equivalent amounts of EV71 at an MOI of 10, and the same concentrations of BmKn2-T5 were then added to wells at 0, 1, 2, 4, 6, and 8 h after infection. At 12 h post-infection, cells from different time points were harvested, and intracellular EV71 RNA and VP1 protein were analyzed using qRT-PCR and Western blotting, respectively.

### 2.8. Plaque Assay

To investigate the effect of scorpion-derived antimicrobial peptide BmKn2-T5 on HSV-1 infection, HSV-1 particles in RD cells were examined using the viral plaque assay. Vero cells were first seeded in a 12-well (2 × 10^5^ cells/well) plate overnight and then cultured with fresh medium when the density of cells reached about 90%. Next, equivalent amounts of EV71 at an MOI of 0.1 and different concentrations of BmKn2-T5 were, respectively, added to the seeded Vero cells. For 1 h incubation at 37 °C with 5% CO_2_, all wells were washed twice with PBS and overlaid with 1 mL of fresh MEM medium containing the same concentration of diluted peptides. After 48 or 72 h incubation to allow for a cytopathic effect, the overlaid medium was replaced with 1 mL of crystal violet staining solution containing 10% formaldehyde. After fixation and staining for 2 h, this solution was removed by rinsing with tap water, and the numbers of plaques in each well were determined by taking photos with a camera and then recording.

### 2.9. Statistical Analysis

All the data were analyzed using GraphPad Prism 8.0 software (La Jolla, CA, USA), and the graphical representations were generated using Adobe Photoshop CS6 (Adobe Systems Inc., San Jose, CA, USA). The significance in the statistical analyses was determined using Student’s T test (*, *p* < 0.05; **, *p* < 0.01; ***, *p* < 0.001), and the data are presented as the means ± SDs.

## 3. Results

### 3.1. BmKn2 and Its Derivative BmKn2-T5 Show Significant Inhibitory Activities against EV71 Infection

Based on our previous study, we selected BmKn2 and five of its derivatives (BmKn2-T1, BmKn2-T2, BmKn2-T3, BmKn2-T4, and BmKn2-T5) as candidates for screening for activity against EV71 infection. BmKn2 isolated from the scorpion *M. martensii* is composed of 13 amino acid residues without disulfide bridges, and its derivatives range from 10 to 12 amino acids in length (Figure 1A). Meanwhile, the amino acid sequence alignment shows there are obvious differences in primary structure, especially between BmKn2-T5 and other peptides. Considering that susceptible cells infected with EV71 facilitate apoptosis and appear to have a cytopathic effect (CPE) through cooperation with viral 3Cpro and 2Apro, a CPE reduction method was used to investigate the inhibitory activity of BmKn2 and its derivatives against EV71 infection [38]. Without peptide treatment, an obvious cytopathic effect was observed for the virus control group, and the number and shape of the cells were lower and rounder, respectively, than those in the cell control group without the virus (Figure 1B). Significantly, BmKn2 and its derivatives can inhibit the EV71-induced CPE in RD cells to varying degrees, and BmKn2 or BmKn2-T5 showed stronger inhibition. Although having only one amino acid residue different from BmKn2-T5, BmKn2-T2 did not exhibit significant antiviral ability in contrast to BmKn2-T5. To further confirm this inhibition, we measured the antiviral effect of BmKn2 and BmKn2-T5 in RD cells by qRT-PCR (Figure 1C,D). Consistently, both BmKn2 and BmKn2-T5 with a concentration of 10 μg/mL showed significant inhibitory activity at the RNA level, indicating that these two peptides have the ability to resist EV71 infection. Together, these results suggest that scorpion venom antimicrobial peptide BmKn2 and its derivative BmKn2-T5 appear to be potent inhibitors of EV71 in vitro.

### 3.2. BmKn2 and BmKn2-T5 Share a Typical Amphiphilic α-Helical Structure

Antimicrobial peptides are small cationic peptides with broad antimicrobial activity that is completely dependent on their structural characteristics and physicochemical properties [39]. To further explore the antiviral mechanism of BmKn2 and its derivative BmKn2-T5, the structural characteristics of the two peptides were first projected using the I-TASSER server and the online HeliQuest server. Both BmKn2 and BmKn2-T5 adopted an obvious α-helical conformation, and the basic residues or the hydrophobic residues were separately gathered on either of the two faces of the α-helix (Figure 2A,B). Compared to BmKn2, which has a hydrophobicity value of 0.538, BmKn2-T5 displays slightly higher hydrophobicity with a hydrophobicity value of 0.600, indicating that both have moderate hydrophobicity and amphiphilicity (Figure 2C,D). Meanwhile, the helical wheel plots for both peptides show they have a similar residue arrangement, which may provide the structural basis for their analogous antiviral activity. To confirm their α-helix structural characteristics, circular dichroism (CD) spectra were recorded using either ddH_2_O or TFE as solvents. As described previously [40], TFE mimics the more hydrophobic environment of bacterial membranes, and, for this reason, it is often used in structural studies related to antimicrobial peptides. Generally, most antimicrobial peptides show a mainly disordered conformation in H_2_O and become highly ordered in the presence of TFE with different concentrations. Both BmKn2 and BmKn2-T5 exhibited an apparent negative peak at approximately 198 nm in ddH_2_O, indicative of a randomly coiled structure in aqueous solution (Figure 2E,F). Meanwhile, a large positive peak at approximately 195 nm and two negative bands at 208 and 222 nm in 30% or 70% TFE solutions were, respectively, observed from the CD spectrum analysis of BmKn2 and BmKn2-T5, indicating that they could form an α-helix-rich structure within the appropriate membrane environment. Taken together, these data suggest that BmKn2 and BmKn2-T5 share the structural characteristics typical of an amphiphilic α-helical structure.

### 3.3. BmKn2-T5 Dose-Dependently Inhibits EV71 Infection at Noncytotoxic Concentrations

To further investigate the antiviral activity of scorpion venom antimicrobial peptide BmKn2 and its derivative BmKn2-T5, the cytotoxicity of the two peptides at various concentrations was first measured using RD cells in an MTS assay. The results show that the 50% cytotoxicity concentration (CC_50_) of BmKn2 in RD cells is about 51.40 μg/mL, while that of BmKn2-T5 peptide in RD cells is about 97.33 μg/mL, indicating that BmKn2-T5 is less toxic than BmKn2 in RD cells (Figure 3A,B). Meanwhile, these results also demonstrate that BmKn2 and its derivatives are substantially noncytotoxic toward RD cells at the concentrations used during antiviral experiments. Because of its low cytotoxicity, we then tested the concentration dependency of BmKn2-T5′s ability to inhibit EV71. We found that BmKn2-T5 can dose-dependently suppress EV71-induced CPE in RD cells, and the cells were almost completely restored by treatment with 10 μg/mL BmKn2-T5 (Figure 3C). Moreover, BmKn2-T5 treatment also resulted in a dose-dependent decrease in EV71 infection at both the RNA and protein levels, for which the 50% inhibitory concentration (IC_50_) against EV71 RNA is 3.61 μg/mL (Figure 3D,E). Taken together, our data show that BmKn2-T5 can significantly inhibit EV71 replication at noncytotoxic concentrations in vitro.

### 3.4. BmKn2-T5 Affects EV71 at the Early Stages of Its Life Cycle

Viruses are highly parasitic, being completely dependent on the energy and metabolic system of their host cells, and they show characteristics typical of living organisms through attachment, entry, replication, assembly, and release of progeny viruses. Many studies into the targets or mechanisms of antiviral agents have mainly focused on the viral life cycle, especially at the early stages, which has proven to be an effective strategy for developing new drugs [37,41]. To further investigate which step in the viral life cycle is utilized by BmKn2-T5 to inhibit EV71, some experiments testing different modes of peptide treatment were designed and performed (Figure 4A). As shown by the results, 10 μg/mL BmKn2-T5 had a slight virucidal effect on EV71 particles at the RNA level when incubated separately with the virus at 37 °C (Figure 4B). Importantly, qRT-PCR and Western blotting analyses demonstrate that BmKn2-T5 exhibits significant inhibitory activity at the early stages of EV71, especially in attachment, where there is 83.8% inhibition, and entry is inhibited 64.4% (Figure 4C,D). To validate this conclusion, we measured the antiviral effect of BmKn2-T5 at different additional time points, and the models of peptide treatment are shown as a schematic diagram in Figure 4E. We observed that BmKn2-T5 consistently induces a significant decrease with treatment for 0 and 1 h, and EV71 RNA was reduced by an average of 94.5% and 43.8%, respectively, at these two time points (Figure 4F). Therefore, these results suggest that BmKn2-T5 inhibits EV71 infection mainly at the early stages of the viral life cycle.

### 3.5. BmKn2-T5 Shows Broad-Spectrum Antiviral Activity against DENV, ZIKV, and HSV-1

As previously reported, EV71 is a non-enveloped single-stranded (+) RNA virus and enters host cells through three stages: attachment, endocytosis, and uncoating [3,42]. Similar to EV71 infection, some enveloped viruses can also enter host cells via the receptor-mediated endocytosis pathway, such as arboviruses represented by Dengue virus (DENV) or Zika virus (ZIKV) and DNA viruses represented by HSV-1 [43,44]. Considering that the mechanism of BmKn2-T5 virus inhibition is likely to be closely linked to the early stages of endocytosis, we further evaluated the antiviral activity of BmKn2-T5 against DENV2, ZIKV, and HSV-1 in Vero cells. Interestingly, BmKn2-T5 treatment also dose-dependently inhibited both DENV2 and ZIKV RNA, and the IC_50_ against these two viruses are, respectively, about 3.36 μg/mL and 1.95 μg/mL, which corresponds to higher inhibitory activity than against EV71 (Figure 5A,B). In addition, we characterized the changes in HSV-1 in response to BmKn2-T5 treatment via plaque assays, and we found that HSV-1 particles also exhibit a dose-dependent reduction with increasing BmKn2-T5 concentration (Figure 5C,D). These data show that BmKn2-T5 can exert significant inhibitory activity against both non-enveloped and enveloped viruses, and the foundation for this broad-spectrum antiviral activity is likely to be related to the endocytosis pathway shared by both types of viruses. To further confirm this conclusion, Vero cells infected with DENV2 or ZIKV were treated with 10 μg/mL BmKn2-T5 at different steps in the viral life cycle. Similarly, BmKn2-T5 showed significant inhibitory activity at the attachment or entry stage of DENV2 and ZIKV but not the replication stage, which suggests that the antiviral mechanism of BmKn2-T5 specifically operates in the early stages of viral infection (Figure 5E,F). These results demonstrate that the scorpion venom antimicrobial peptide derivative BmKn2-T5 exerts broad-spectrum antiviral effects by affecting the early stage of the viral life cycle.

## 4. Discussion

As one of the causative pathogens of HFMD, EV71 may be associated with fatal neurological syndromes, posing a serious public health threat. However, there are few antiviral drugs approved for broadly preventing EV71 infection in clinics [1]. Although several small molecules targeting each step of the virus life cycle have been extensively tested and developed in recent years, it is necessary to find innovative and effective antiviral drugs against EV71 infection [15,45]. Previously, numerous antimicrobial peptides derived from scorpions have been identified and shown to possess potential pharmacological activities in pathogen invasion, and the antiviral activity of scorpion-derived AMPs is still an area worthy of further exploration, particularly in use against non-enveloped viruses [22,46]. In this study, we selected BmKn2, characterized by the scorpion *M. martensii*, and five of its derivatives as candidates in screening to investigate their inhibitory activity against EV71 infection. Interestingly, only BmKn2 and BmKn2-T5 can significantly inhibit EV71-induced CPE in RD cells, and not BmKn2-T1, BmKn2-T2, BmKn2-T3, nor BmKn2-T4. Based on the amino acid sequence alignment of BmKn2 and BmKn2-T1, we hypothesized that the third glycine of BmKn2 may be the key active site for its antiviral activity. Similarly, the fifth leucine of BmKn2 also appears important based on the sequence alignment of BmKn2-T2 and BmKn2-T5. Therefore, further work examining the effects of key amino acid residues on the antiviral ability of the two peptides is needed. We also investigated the inhibitory activity of other scorpion-derived antimicrobial peptides against EV71 infection using a CPE reduction method. Compared to BmKn2 and its derivative BmKn2-T5, other scorpion-derived AMPs were not found to exhibit significant antiviral ability against EV71 infection. These data indicate that scorpion-derived antimicrobial peptides with different structures or properties show various pharmacological activities in EV71 infection.

Based on structural characteristics analysis, we found the two antiviral peptides have a typical amphiphilic α-helical structure, which serves as the structural basis for their possible action on cell membrane or virus-associated membrane transfer processes. Several scorpion venom-derived peptides have been confirmed to exert significant antiviral activities, and most were discovered by our research group, including Kn2-7 in acting against HIV-1 infection [33]. To further confirm the antiviral activity of BmKn2 and BmKn2-T5, we first compared the cytotoxicity of two peptides and found that BmKn2-T5 is less toxic than BmKn2 in RD cells. Previous studies have reported that the length and hydrophobic residues of peptides are important for their cytotoxicity, where BmKn2-T5 has favorable structural characteristics and safety, making it an ideal candidate for pharmacological applications [47]. We then demonstrated that BmKn2-T5 treatment results in a dose-dependent decrease in EV71 infection at noncytotoxic concentrations, indicating that it may be a potential candidate for development as an antiviral drug. Recently, cathelicidin antimicrobial peptides LL-37 and CRAMP were reported to inhibit EV71 infection via regulating antiviral response and inhibiting viral binding, and two cathelicidin-derived peptides, LL-18 and FF-18, showed higher inhibitory activity against EV71 through blocking virus–receptor interactions and inhibiting viral uncoating [13,48]. These studies further suggested that antimicrobial peptides might be excellent candidates for anti-EV71 drugs and proposed potential targets or mechanisms for antimicrobial peptides against EV71.

Consequently, time-of-drug-addition experiments were designed and performed to investigate which step in the viral life cycle is utilized by BmKn2-T5 to inhibit EV71. Importantly, it was demonstrated that BmKn2-T5 exhibits significant inhibitory activity at the early stages of EV71, especially in attachment and entry. This phenomenon is different from the action stages of most scorpion venom-related antiviral peptides reported in the literature, such as Ctry2459, Hp1090, Hp1036, Hp1239, Kn2-7, and mucroporin-M1, which exert antiviral activity by inactivating infectious viral particles or acting at post-entry stages [23,49]. These data suggest that BmKn2-T5 may affect the early stages of EV71 via a novel antiviral mechanism. For example, numerous receptors/co-receptors anchored on the host cell membrane are being continuously identified, including scavenger receptor B2 (SCARB2), P-selectin glycoprotein ligand-1 (PSGL-1), heparan sulfate, and annexin II (Anx2), and they may interact with BmKn2-T5 to regulation EV71 infection [3,50]. Given that the virus inhibition mechanism of BmKn2-T5 is likely to be closely linked to the early endocytosis stage, we further evaluated the antiviral activity of BmKn2-T5 against enveloped viruses, such as arboviruses and DNA viruses. Interestingly, BmKn2-T5 also showed significant and dose-dependent inhibitory activity against DENV, ZIKV, and HSV-1, particularly in the attachment or entry stage. These data show that BmKn2-T5 can exert broad-spectrum inhibitory activity against both non-enveloped and enveloped viruses, and the antiviral activity against both is likely due to the mechanism being based on the endocytosis pathway, which is common to both. Certainly, further work on specific antiviral targets or mechanisms is warranted.

## 5. Conclusions

In this study, we found that scorpion venom antimicrobial peptide BmKn2 and its derivative BmKn2-T5, both sharing a typical amphiphilic α-helical structure, exhibit significant inhibitory activity against EV71. Compared with BmKn2, BmKn2-T5 exhibits lower cytotoxicity in RD cells and dose-dependently inhibits EV71 infection at noncytotoxic concentrations. In addition, it was revealed that BmKn2-T5 mainly inhibits EV71 infection at the early stage, but not its virion or replication. Finally, we demonstrated that BmKn2-T5 surprisingly has broad-spectrum antiviral activity against both non-enveloped and enveloped viruses. Although BmKn2-T5 has been identified as the primary active component to act against the virus in vitro, its structural organization, toxicity, and stability remain to be developed for clinical purposes. Based on this effective molecular probe or scaffold, further understanding of its structural and functional properties, and the design of pharmacologically active peptides with high activity and selectivity are important focuses for the future. Taken together, our results demonstrate that BmKn2-T5, a scorpion venom antimicrobial peptide derivative, exerts broad-spectrum antiviral effects by affecting the early stages of the viral life cycle, which may lead to the discovery of new antiviral candidates for pharmacological applications.

## Figures and Tables

**Figure 1 biomolecules-14-00545-f001:**
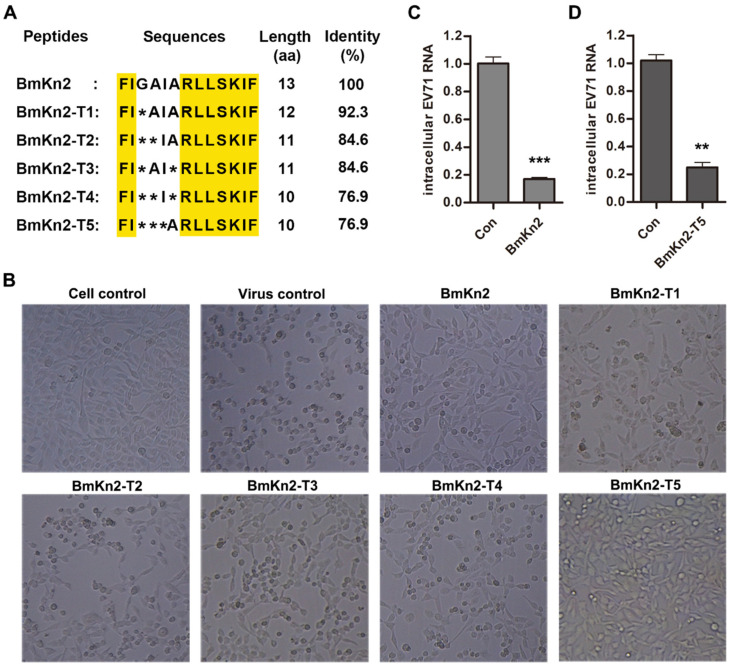
Screening of scorpion venom antimicrobial peptide BmKn2 and its derivatives against EV71. (**A**) Multiple alignment of BmKn2 and five derivatives. Conserved amino acids are highlighted in yellow. (**B**) Inhibitory activity of BmKn2 and its five derivatives against EV71-induced CPE in RD cells. RD cells infected with EV71 at an MOI of 1 were treated with 10 μg/mL BmKn2 or its derivatives, and the morphology of RD cells was recorded at a magnification of 100× after 24 h post-infection. (**C**) Inhibitory activity of BmKn2 against EV71 assessed by qRT-PCR. (**D**) Inhibitory activity of BmKn2-T5 against EV71 assessed by qRT-PCR. Data are presented as the means ± SD from three independent experiments (*, *p* < 0.05; **, *p* < 0.01; ***, *p* < 0.001).

**Figure 2 biomolecules-14-00545-f002:**
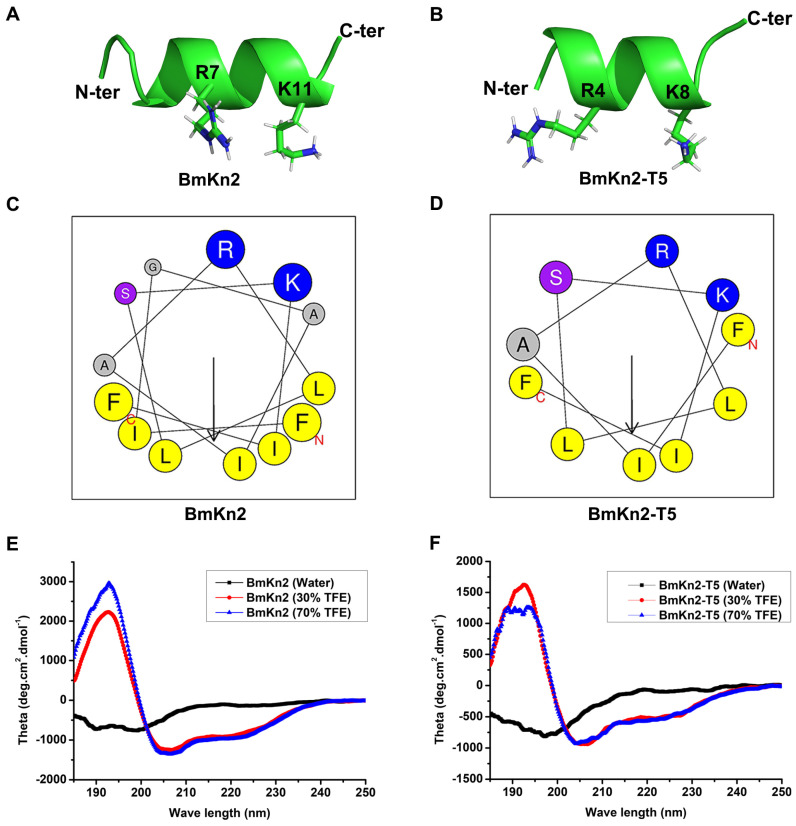
Analysis of BmKn2 and BmKn2-T5 structural characteristics. (**A**,**B**) Three-dimensional structures of BmKn2 and BmKn2-T5 obtained using I-TASSER. (**C**,**D**) Helical wheel plots of BmKn2 and BmKn2-T5 obtained using the HeliQuest server. In the helical wheel plots, arrows represent the direction of summed vectors of hydrophobicity, and residues marked in blue and yellow represent positively charged alkaline and hydrophobic amino acids, respectively. Meanwhile, the residue marked in purple represents Ser, and residues marked in gray represent Gly and Ala. (**E**,**F**) CD spectrum analysis of BmKn2 and BmKn2-T5. The CD spectra were recorded in water, 30% TFE, and 70% TFE solutions, respectively. N-ter, N-terminus; C-ter, C-terminus.

**Figure 3 biomolecules-14-00545-f003:**
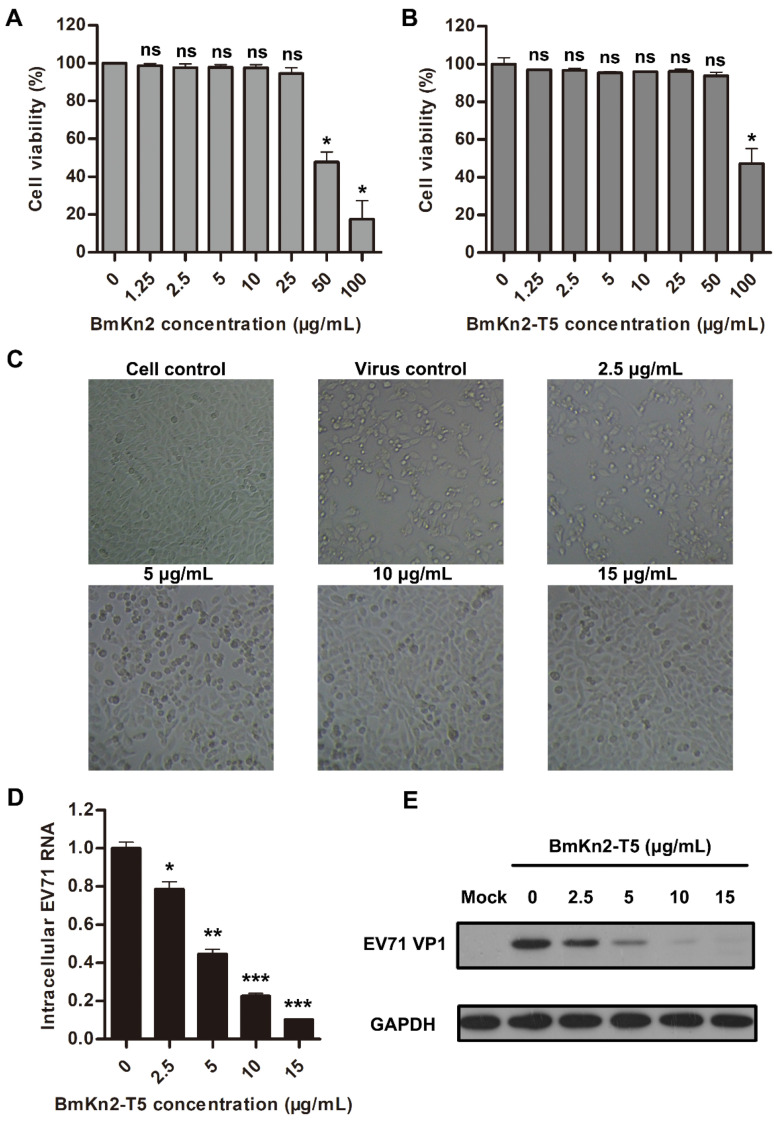
Dose-dependent antiviral effect of BmKn2-T5 on EV71 in RD cells. Cytotoxicity of (**A**) BmKn2 and (**B**) BmKn2-T5 in RD cells. The cell viability of RD cells treated with different concentrations of BmKn2 and BmKn2-T5 was measured using an MTS assay. (**C**) Inhibitory activity of BmKn2-T5 with different concentrations to EV71-induced CPE in RD cells. RD cells infected with EV71 at an MOI of 1 were treated with different concentrations of BmKn2-T5, and the morphology of RD cells was recorded at a magnification of 100× at 24 h post-infection. Dose-dependent inhibitory effect of BmKn2-T5 on EV71 analyzed using (**D**) qRT-PCR and (**E**) Western blotting. Data are presented as the means ± SD from three independent experiments (ns, not significant; *, *p* < 0.05; **, *p* < 0.01; ***, *p* < 0.001). Original images of (**E**) can be found in Appendix A.

**Figure 4 biomolecules-14-00545-f004:**
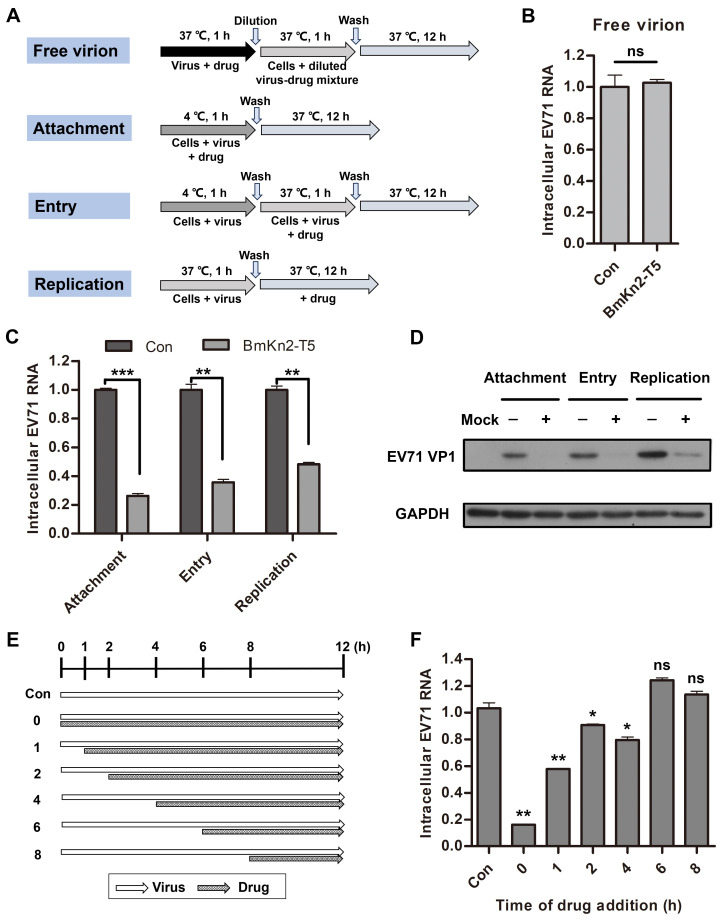
Effect of BmKn2-T5 on different stages of EV71 life cycle. (**A**) Schematic diagram for studying the action stage of BmKn2-T5 on EV71. (**B**) Effect of BmKn2-T5 on EV71 free virion in RD cells. EV71 attachment, entry, and replication in RD cells detected using (**C**) qRT-PCR and (**D**) Western blotting. (**E**) Schematic diagram of BmKn2-T5 treatment for different time points on EV71 infection. (**F**) Effect of BmKn2-T5 added for different time points on EV71 infection. RD cells infected with EV71 were incubated with 10 μg/mL BmKn2-T5 for the indicated times, and intracellular EV71 RNA was analyzed using qRT-PCR at 12 h post-infection, respectively. Data are presented as the means ± SD from three independent experiments (ns, not significant; *, *p* < 0.05; **, *p* < 0.01; ***, *p* < 0.001). Original images of (**D**) can be found in Appendix A.

**Figure 5 biomolecules-14-00545-f005:**
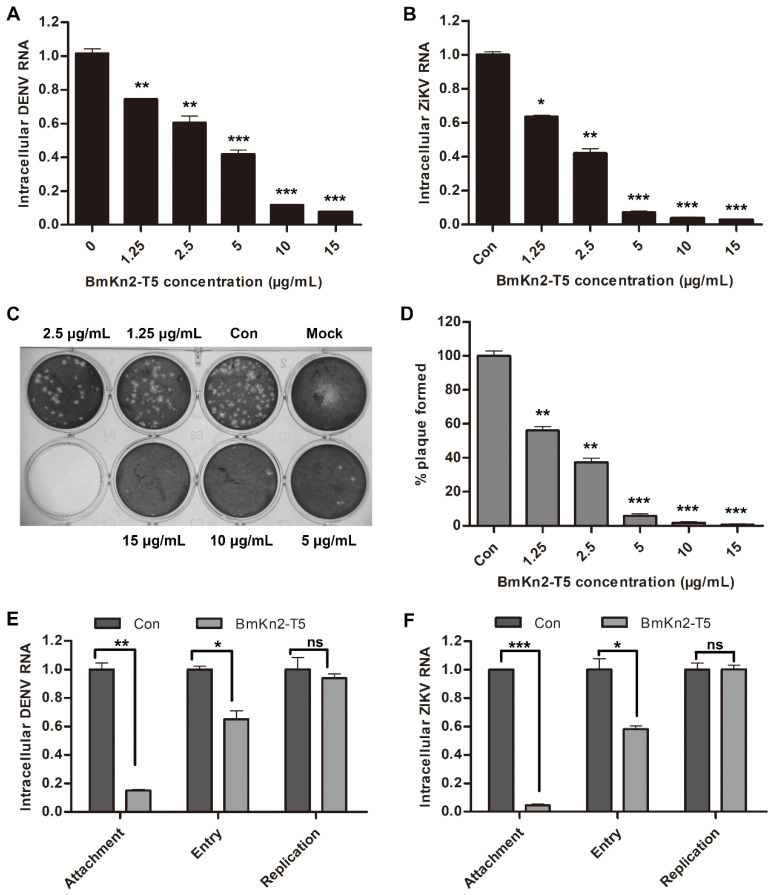
Antiviral activities of BmKn2-T5 against DENV, ZIKV, and HSV-1 at noncytotoxic concentrations. Inhibitory effect of BmKn2-T5 on (**A**) DENV2 and (**B**) ZIKV. Vero cells infected with ZIKV at MOI of 0.1 were treated with different concentrations of BmKn2-T5 for 24 h, and the level of intracellular DENV2 or ZIKV RNA was analyzed using qRT-PCR. (**C**,**D**) Inhibitory effect of BmKn2-T5 on HSV-1 infection in vitro. Vero cells infected with HSV-1 were treated with different concentrations of BmKn2-T5 for 1 h, and intracellular HSV-1 was then detected using plaque assays. Meanwhile, plaques were quantified by counting, and the results are described in percentages for the statistical analysis. (**E**,**F**) Effect of BmKn2-T5 on DENV or ZIKV attachment, entry, and replication in RD cells detected using qRT-PCR. Data are presented as the means ± SD from three independent experiments (ns, not significant; *, *p* < 0.05; **, *p* < 0.01; ***, *p* < 0.001).

**Table 1 biomolecules-14-00545-t001:** Primers used in this study.

Name	Direction	Sequence (5′-3′)
hGAPDH	Sense	TGATGACATCAAGAAGGTGGTGAAG
	Antisense	TCCTTGGAGGCCATGTGGGCCAT
EV71	Sense	TGAATGCCGGCTAATCCCAACT
	Antisense	AAGAAACACGGACACCCAAAG
DENV	Sense	GGCCTCGACTTCAATGAGATGG
	Antisense	CCTGTTTCTTTGCATGGGGAT
ZIKV	Sense	TTGTGGAAGGTATGTCAGGTG
	Antisense	ATCTTACCTCCGCCATGTTG

## Data Availability

All data are presented in this paper.

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
