# Peer review of "Scorpion Venom Antimicrobial Peptide Derivative BmKn2-T5 Inhibits Enterovirus 71 in the Early Stages of the Viral Life Cycle In Vitro"

_biomolecules, 2024, doi:10.3390/biom14050545_

Round 1

Reviewer 1 Report

Comments and Suggestions for Authors

See attached file.

Comments on the Quality of English Language

Author Response

Thank you very much for taking the time to review this manuscript. Please find the detailed responses below and the corresponding revisions/in track changes in the re-submitted files.

Reviewer 2 Report

Comments and Suggestions for Authors

The manuscript entitled “Scorpion venom antimicrobial peptide derivative BmKn2-T5 2

inhibits the early stages of EV71 life cycle in vitro” authored by Zhiqiang Xia et al presents results pointing out the in vitro antiviral properties of scorpion-derived antimicrobial peptide BmKn2 and its derivatives

The topic is very important and of great interest to the medical field.

The introduction is well structured and documents the selected viral pathogen pathogenic role, the associated disease geographical distribution, consequences and available control and preventive measures, as well as the antimicrobial peptides presentation.

The antiviral properties are evaluated using a complex protocol, still the information provided in the content of the abstract and the introduction do not entirely match the content of the results, discussion and conclusion. If initially, the authors refer to the antiviral properties against EV71, the text contains information regarding the in vitro antiviral efficacy against other viruses.

The authors should solve these confusing aspects with text modifications of the abstract, otherwise, the conclusion of the abstract is an overstatement.

Thus, for the abstract:

-        the authors are recommended to remove the marked text, rephrase all the sentences of the abstract (too many details, word repetitions), include the investigation protocol – concise text presenting methods,  add “in vitro” as all the results refer to in vitro demonstrated properties.

As a typical representative of unenveloped RNA viruses, enterovirus 71 (EV71) is the main

pathogenic factor of hand-foot-and-mouth disease (HFMD) in infants, which causes serious threats to human health and life worldwide, especially in the Asia-Pacific region.

might serve as a broad-spectrum antiviral agent. Our study may lead to the discovery of a new candidate against viruses.

-        “Our study may lead to the discovery of a new candidate against viruses. “ this phrase should be modified, it could be an overstatement and the authors could use a better formulation eg the investigated peptides proved valuable in vitro antiviral properties against EV71, with a notable effect against …, but also against ” or furthermore, the antiviral properties were found against …using the method …

Moderate editing of the English language is required for the whole manuscript to improve the syntax and formulations, eg :

Lines 39- 41 “….in 1969 is mainly infected in infants and young …” could be rephrased as:

Firstly isolated from meningitis patients in California in 1969, EV71 is the main pathogen of hand, foot and mouth disease (HFMD), mainly affecting infants and children under 6 years of age

Line 50 is still an of utmost importance [11]. 

Comments on the Quality of English Language

Moderate editing of the English language is required for the whole manuscript to improve the syntax and formulations

Author Response

(The authors gave the same response as above.)

Reviewer 3 Report

Comments and Suggestions for Authors

In this article, Xia et al. studied the scorpion venom antimicrobial peptide derivative BmKn2-T5 and its inhibition of the early stages of the EV71 life cycle in vitro. Although many peptides isolated from animals focus on enveloped viruses, the authors specifically investigated the scorpion-derived peptide, which is novel in the literature.

After testing the peptide and its five derivatives, both BmKn2 and the derivative BmKn2-T5 exhibited a significant inhibitory effect on EV71. Although both peptides displayed secondary structure characteristics with a typical amphiphilic α-helical structure, BmKn2-T5 showed lower cellular cytotoxicity than BmKn2. Additionally, BmKn2-T5 was further identified to inhibit EV71 in a dose-dependent manner. The authors also demonstrated that BmKn2-T5 suppressed the replication of enveloped viruses including DENV, ZIKV, and HSV-1, suggesting that BmKn2-T5 might serve as a broad-spectrum antiviral agent.

The overall study is well-planned and conducted, and the results are important for the field since the peptide and its derivatives are novel compounds as antiviral agents.

Before acceptance, the following points need to be addressed:

  • The authors provided a thorough explanation of EV71 and its threat to human health. However, the introduction lacks discussion on current studies against it. For example, what are the other recent studies being conducted in this area?
  • It would be very helpful to include a figure showing the potential mechanism of the peptide on the virus in the introduction section.
  • While the results are presented perfectly, the discussion could be improved by adding more existing data regarding other candidate molecules tested for the same activity.
  • It appears that the authors are experts on this molecule, and this study was conducted after showing activity against gram-positive and gram-negative pathogens. It would be beneficial to mention the next studies in the conclusion section, outlining future directions for the audience.

Author Response

(The authors gave the same response as above.)

Round 2

Reviewer 1 Report

Comments and Suggestions for Authors

Please, indicate in the Section 2.4. the width of the cell used to measure CD-spectra and provide concentration of the peptides in the cell during measurement. Also, describe whether baseline correction of the spectra was performed and if yes, how.

Figure 2,e. Mdeg is raw value. You calculated mean residue ellipticity. Please, change Dimensionality of Y-axis from Mdeg to Mean residue ellpticity. 

English improvements are still required (see further).

Comments on the Quality of English Language

young children (simply children, because age is indicated, 6 y.o), usage of articles.

Author Response

Thank you very much for taking the time to review this manuscript. Please find the detailed responses below and the corresponding revisions/ in track changes in the re-submitted files.
